# FTO and Anthropometrics: The Role of Modifiable Factors

**DOI:** 10.3390/jfmk7040090

**Published:** 2022-10-18

**Authors:** Cassandra Evans, Jason Curtis, Jose Antonio

**Affiliations:** 1Health and Human Performance, Nova Southeastern University, Davie, FL 33314, USA; 2Healthy Sciences, Rocky Mountain University of Health Professions, Provo, UT 84606, USA; 3Exercise Science, Keiser University, West Palm Beach, FL 33411, USA

**Keywords:** genomics, physical activity, nutrition

## Abstract

Numerous gene variants are linked to an individual’s propensity to become overweight or obese. The most commonly studied gene variant is the FTO single nucleotide polymorphism. The FTO risk allele is linked with increased body mass, BMI and other lifestyle factors that may perpetuate an individual’s risk for obesity. Studies assessing eating behaviors, eating preferences, nutrition interventions and other lifestyle factors were reviewed. These studies demonstrated a clear difference in eating behaviors and preferences. Lifestyle modifications including physical activity and diet were effective in weight management even in those with the risk allele.

## 1. Introduction

The obesity epidemic affects 650 million people worldwide [1]. The pathogenesis of obesity is multifaceted. Factors such as activity level, socioeconomic status, environment and genetics contribute to an individual’s likelihood of becoming obese. Despite the numerous health initiatives implemented, obesity rates and related comorbidities continue to rise [1]. Developing a deeper understanding of the complex relationship between the external environment and gene expression could prove to be powerful tool.

Genome-wide associated studies (GWAS) have identified more than 300 single nucleotide polymorphisms (SNPs) associated with obesity. The SNP with the most significant association with obesity among numerous populations is FTO Alpha-Ketoglutarate Dependent Dioxygenase (FTO) [2,3]. Individuals with SNPs occurring on intron-1 of the FTO gene (rs9939609 (A/-), rs17817449 (G/-), rs3751812 and rs1421085 (C/-)) are more likely to suffer from obesity (Table 1) [2,3,4,5,6]. There are multiple mechanisms that alter phenotypes without causing permanent changes in DNA sequence. These include methylation, acetylation and phosphorylation, all of which can influence gene expression [7]. Early studies first associated FTO SNPs with a greater risk of type 2 diabetes [3]. As research in this area developed, it became clear the increased risk of type 2 diabetes was due to higher BMI or excess weight [3]. Obesity-related traits correlating to the FTO loci can be observed in Western European, Hispanic/Latino, Asian and Pima Indian populations [3,8]. Genotype testing revealed that three SNPs (rs17817449, rs3751812 and rs1421085) were strongly associated with obesity (class III) in French individuals with European descent [6]. Similar associations were observed in French children and Swiss individuals [6]. In African American populations [3], this correlation is much weaker.

Andreasen et al. conducted a large population study observing metabolic traits and between FTO variations. Subjects had type 2 diabetes (n = 3856). Subjects without type 2 diabetes (n = 4861) served as the control group. Blood samples were used to assess genotype; questionnaires assessed physical activity; and anthropometrics were assessed via BMI, weight and waist circumference. Subjects with the risk allele from both groups weighed 3.3 kg more than those without the risk allele. BMI and waist circumference was greater; however, when circumference was adjusted for BMI, differences were not significant. These results suggest that a higher BMI is correlated with greater fat mass overall rather than greater abdominal fat mass [9].

Antonio et al. (2018) assessed the genotype and anthropometric measures in an exercise trained populations. Body composition was assessed using dual-energy X-ray absorptiometry (DXA), and saliva samples were used to determine genotype. Only subjects with a C/- risk allele and those with non-risk allele (TT) were compared. No significant differences in body weight were reported between groups [24]. The risk group (C/-) had significantly higher fat mass and body fat percentage. This study clearly indicates that differences in body composition are still present in individuals with risk alleles irrespective of weight status. Despite the advances in genomic research, the mechanistic action in which FTO SNPs contribute to obesity is not fully understood. It has been suggested that factors such as eating behaviors, energy intake and responsiveness to lifestyle interventions regulate FTO expression.

Efforts to treat obesity and related health issues have been rather ineffective. The “one size fits all” approach is not applicable in terms of nutrition and exercise. Lifestyle interventions should be personalized and based on the individual’s preferences, goals and current lifestyle. Is genetic testing the new way to personalize lifestyle interventions? Is there enough evidence to support using an individual’s DNA to make lifestyle recommendations? The aim of this paper is to identify the implications of various FTO SNPs on dietary habits and to examine effectiveness of lifestyle interventions.

## 2. Methods

In the present review, we used studies using exercise or/and diet interventions and those investigating dietary habits/preferences related to the FTO allele. We focused on treatment techniques that involved aerobic training, resistance training or both trainings, as well as diet interventions or nutrition analysis. We also selected studies that assessed FTO status and any FTO risk allele.

An electronic search was performed using the databases PubMed and Google Scholar using the following search terms for each individual database:“FTO” OR “diet” OR “body composition”;“FTO” OR “satiety” OR “dietary intake”;“FTO” OR “eating preferences” OR “weight loss”;“FTO” OR “weight gain” OR “metabolic health”;“diet” OR “nutrition” OR “protein” OR “FTO” OR “lifestyle modification” OR “FTO”;“FTO” or “physical activity”.

Recent literature published after 2000 was considered. We analyzed both titles and abstracts to decide whether a study could be included in our review. Articles were included if they fulfilled the following criteria: peer-reviewed, human subjects only and/or research-based.

## 3. FTO and Energy Intake

The FTO SNP is thought to be involved with the regulation of energy intake. Numerous studies have reported individuals with risk alleles to have a higher energy intake, specifically from fats and sugars [10,11,12,15,16,17,18,19,20,22,23,25,26]. Those with the risk allele experience lower levels of satiety and reported poor eating behaviors [10,11,12,15,16,17,18,19,20,22,23,25,26]. Harbron et al. studied FTO variants (C-allele, G- allele and T- allele) and their relationship to dietary intake, eating behavior, physical activity and psychological health. Subjects, ages 25–40 years old, were overweight or obese (BMI ≥ 27 kg/m^2^) and Caucasian. Food Frequency questionnaires (FFQ) were used to assess dietary intake, and eating behaviors were assessed using the validate Three-Factor Eating questionnaire (TFEQ). Blood tests were used to determine subject’s genotype. There were a greater number of subjects with one of the risk alleles than those with a non-risk allele. The results of the FFQ indicated higher intakes of saturated fat and refined sugars in the GG allele group when compared to non-risk group [22]. TFEQ revealed that perceived hunger scores were significantly greater in subjects with one of the risk alleles. Subjects with two risk alleles (homozygous) scored 2.86 times higher than those with one risk allele in the perceived hunger category. Additionally, subjects with C-alleles scored highest on internal locus for hunger control. These results demonstrate a clear difference in eating behaviors between risk and non-risk alleles. The authors suggest that the persistent feeling of hunger contribute to an increase in eating “just to eat”, thus contributing to a higher energy intake [22].

McCaffery et al. conducted a similar study that found an association between risk alleles and dietary intake. Subjects were recruited from the clinical trial Look AHEAD (Action for Health in Diabetes), which was initially designed to study the effects of various lifestyle interventions on cardiovascular disease risk [26]. Subjects (n = 2075) were obese and had type 2 diabetes. Unlike Harbron’s study, subjects were culturally diverse. Blood samples were used to assess genotype. FFQ was used to assess dietary intake and number of eating episodes. The results indicated a greater number of eating episodes and a modest increase in daily energy and fat intake associated with FTO risk alleles. Following statistical adjustment for total energy intake, significant differences in eating episodes remained [23]. The propensity to consume higher fat foods and more meals or snacks throughout the day likely contributes to the higher energy intake.

A more recent study looked at the mechanisms related to satiety and FTO risk allele (AA). This is one of the few studies where all subjects were not overweight or obese; however, the average BMI was categorized as obesity class 1. Food behaviors were assessed via TFEQ. Blood samples were collected to determine genotype, plasma glucose, and hunger hormones ghrelin and leptin. Eating habits via caloric intake at a buffet and brain response to visual cues was evaluated. In line with the previously mentioned studies, subjects with the risk allele consumed more calories than their non-risk counterpart. TFEQ revealed less feelings of fullness and attributing greater appeal to high fat foods in risk allele subjects. Interestingly, premeal and post meal testing observed greater brain activity in response to fattening visual cues in the risk allele group. No difference in hormones were observed [16]. There are many more studies which consistently report correlations between energy intake, differences in dietary intake, eating behaviors and satiety [11,13,16,19,20,22,23]. A limitation of these studies is the weight status of subject. These studies tend to recruit overweight or obese subjects. It is not clear if these differences contributed to subject’s obesity or are the result of poor dietary habits stemming from being obese. There is evidence to suggest hedonistic eating tendencies in obese individuals independent of their genotype [25].

Numerous studies that have observed similar eating behaviors in children and adolescents [10,11,12,15,17,18,19]. Rivas et al. conducted a study measuring eating behaviors and FTO variations. Subjects’(n = 122) ages ranged from 8 to 14 years old. The Child Eating Behavior questionnaire and TFEQ were used to evaluate eating behaviors. The Food Reinforcement Value questionnaire was administered to assess reinforcement value of food in relation to their eating behavior [15]. Children were told they would receive all rewards and then asked to choose between food and non-food (stickers) rewards. Male subjects with the FTO risk allele scored lower in measures of satiety and higher in measures of food responsiveness and emotional eating. The authors did not explain the sex-specific difference observed in their study [15]. These findings are similar to other FTO studies observing eating behaviors and children [10,12,17,18]. Both Wardle et al. (2008) and Emond et al. reported lower levels of satiety in children with FTO risk alleles. Adolescents with the risk allele (AT and AA) were more likely to be overweight or obese. The high-risk allele group (AA) scored the lowest on feelings of satiety. Wardle et al. (2008) found that children with the risk alleles (AA and AT) reported lower levels of satiety and had a higher energy intake. In a later study, Wardle et al. (2009) observed higher energy intakes in children with risk allele. During the later study, children were offered biscuits (two sweet and one savory) 1 h after consuming a full meal. They were instructed to eat as many they wanted. Both risk allele groups (AA and AT) consumed more biscuits [17]. This is a clear example of eating in the absence of hunger suggesting that individuals that carry the risk alleles may not feel satiety or have low levels of eating control.

Similarities in energy intake and eating habits are present in non-obese children with the risk allele. Ranzenhofer et al. observed higher energy intake in the risk allele groups. This study found that high energy intakes were more likely to occur in Caucasian subjects with the risk alleles. These findings are consistent with previous studies that report African Americans with risk alleles are less likely to exhibit obesogenic behaviors [3]. Although satiety was not evaluated, children were given 60 min to consume as much food as they wanted at each meal [18]. Based on previous studies suggesting lower levels of satiety, it can be inferred that perceived fullness, or a lack of, contributed to children with risk alleles consuming more at each meal.

The literature surrounding energy intake and eating behaviors in children and adult unequivocally demonstrates distinct differences in individuals with carrying risk allele and those with no risk (TT). These differences can be detected early on in children irrespective of current weight status. As mentioned previously, simply carrying a risk allele does not dictate obesity. However, the combination of increased energy intake, lower levels of satiety and an obesogenic environment contribute to weight gain and health status of those with a higher risk. Please refer to Table 2 for a summary of studies.

## 4. FTO and Physical Activity

There is a plethora of literature highlighting the role that physical activity plays in health. Regular physical activity is linked with lower incidence of chronic diseases, improved cardiorespiratory fitness and improved mental health. Increasing physical activity is one modality for addressing obesity. Given the multitude of benefits of physical activity, some have suggested physical activity, or a lack of, can influence the expression of the FTO risk allele [9,14,21,29].

Andreasen et al. [9] used questionnaires to measure physical activity. A correlation between physical inactivity and greater weight was reported. This correlation was not observed between sedentary individuals with the non-risk allele (TT) [9]. This study did not measure differences between physical activity habits of non-risk alleles to risk alleles. These results suggest that physical inactivity may be more detrimental to those with the risk allele. Kim et al. [14] studied the interaction of physical activity with measures of weight and BMI in FTO variations. Subjects who were regularly engaged in physical activity had lower BMI and weight. Although not significant, sedentary subjects with the risk allele were reported to have a higher BMI and weight [14]. One study assessed the physical activity levels of individuals with risk alleles compared to those without risk. Hubacek et al. [21] used questionaries and available DNA information from individuals participating in the HAPIEE (Health, Alcohol and Psychosocial factors In Eastern Europe) project. This identified the rs17817449 or G-allele as the risk variation. When compared to the non-risk group (TT), no differences in hours pf physical activity performed each week were reported [21].

Based on the findings of the aforementioned studies, FTO SNPs do not appear to influence physical activity. These studies demonstrate that lack of physical activity could result in weight gain; however, this is true for all populations. Sedentary lifestyles are associated with a high incidence of comorbidities and an increased likelihood of being overweight or obese. Please refer to Table 3 for a summary of studies.

## 5. FTO and Nutrition Interventions

It has been suggested that differences in eating behaviors are major contributing factors to becoming overweight and obese. There is limited evidence that suggests individuals with one of the risk variants respond differently to lifestyle interventions such as diet, macronutrient intake and physical activity. One study examined changes in weight and body composition of individuals following a Mediterranean diet for 3 years [40]. Subjects were recruited from previous conducted clinical trial examining the effects of the Mediterranean diet [41]. Anthropometric measurements and genotype were measured at baseline. Dietary intake was assessed using a food frequency questionnaire. Anthropometric measurements were reassessed at the end of 3 years. Surprisingly, subjects with the risk allele (A/-) experienced less weight regain than non-risk subjects despite have higher baseline weight and BMI. Razquin et al. [40] suggest the macronutrient distribution associated with the Mediterranean diet may interact with the A-allele.

In one of the few studies conducted using exercise trained individuals, Antonio et al. (2019) [34] assessed genotype and the effects of a short-term hypocaloric diet. Subjects (n = 47) were all exercise trained and grouped together based on their risk type (C-, A- or G-). Body composition was measured using a DXA. MyFitnessPal was used to track intake and saliva samples were collected to determine genotype. Subjects were instructed to reduce their total intake by 20–25% and maintain a higher protein intake (~ 2.0 g/kg/d) for 4 weeks. Both groups lost a significant amount of weight. The risk group experienced a significant reduction in body fat percentage when compared to the non-risk group. Although a short-term study, the results suggest that a reduction in total energy intake and high protein diet can be effective for those with carrying a risk allele [34]. Likewise, Zhang et al. [27] observed beneficial effects of a high protein diet in individuals with the risk allele. Obese subjects (n = 742) were randomly assigned to 1 of 4 diets (Diet 1: 20% Fat, 15% Protein and 65% Carbohydrate; Diet 2: 20% Fat, 25% Protein and 55% Carbohydrate; Diet 3: 40% Fat, 15% Protein and 45% Carbohydrate; and Diet 4: 40% Fat, 25% Protein and 35% Carbohydrate). Anthropometrics and genotype were assessed. Of the 80% of subjects that completed the two-year study, those with the A-allele experienced greater weight loss when following a high protein diet. Additionally, the risk allele group following a high protein diet experienced a greater loss in fat mass and fat free mass. This was not observed in the risk group following a low protein diet [27].

Haupt et al. [30] conducted a study on a subset of subjects participating in a larger longitudinal study. Over the course of 9 months, subjects were asked to reduce their intake via fat by 30%, to increase their fiber intake and to exercise at least 3 h each week. Body composition was assessed using MRI, and all subjects were genotyped. No significant differences in changes of fat deposit and weight loss were reported between risk and non-risk allele carriers.

Another study evaluated the differences in weight loss and adipocytokine levels between two hypocaloric diets [31]. Both diets provided approximately 1500 kcal; diet 1 was labeled low-carbohydrate (38% carbohydrates, 26% proteins and 36% fats), and diet 1 low-fat (53% carbohydrates, 20% proteins and 27% fats). Subjects were instructed to adhere to diet for 3 months and to participate in 3 h of physical activity each week. No significant associations between risk allele and weight loss were reported. Lower levels of leptin were reported in the risk allele group following the low fat diet.

Verhoef et al. [32] examined multiple risk alleles including FTO rs9939609. Subjects followed a low-calories diet (50 g carbohydrates, 52 g protein and 7 g fat) for 8 weeks and then transitioned to a weight maintenance plan. No statistical differences were reported between weight loss in risk alleles, despite having higher BMI at baseline. Rauhio et al. [33] conducted a similar study starting with 3 months of a very low energy diet followed by 6 months of weight maintenance. All subjects were female and obese. The findings of this study were consistent with those of Verhoef et al. and others.

These studies suggest that carrying the FTO SNP does not interfere with the ability to lose weight when following a hypocaloric diet. Moreover, it appears that certain macronutrient distributions, high protein or low fat promote favorable changes in body composition and promote the maintenance of weight loss. Please refer to Table 3 for a summary of studies.

## 6. FTO and Exercise Interventions

Unlike energy intake and eating behaviors, FTO SNPs do not influence an individual’s tendency to engage in physical exercise [9,14,21]. There are a few studies that examined the effects of exercise type on FTO gene expression and response to exercise program. Raniken et al. prescribed an exercise training program based on individual heart rate and VO_2_Max. Subjects were sedentary prior to the study, and all training sessions were supervised. No statistical differences in weight loss anthropometrics were reported between risk and non-risk carriers. Interestingly, Caucasian subjects with C/C homozygotes experiences greater changes in body fat percentage than those with A/A homozygotes. These changes were not observed in black subjects.

Leońska-Duniec et al. [36] conducted a study observing the effects of a 12-week training program in Polish women. The training program consisted of low and high-impact exercises. At baseline and throughput the study, risk allele carriers had higher BMI vales. Both groups saw improvements in anthropometrics and metabolic parameters suggesting that physical activity can improve overall health irrespective of FTO status. Mitchell et al. also observed differences in subjects with A/A homozygotes (risk) and C/C homozygotes (non-risk). Subjects were randomly assigned to one of four groups: control and three exercise groups adhering to 50%, 100% and 150% of the NIH Consensus Development Panel recommendations for physical activity. All subject experienced improvements in cardiorespiratory fitness and loss weight. A/A carriers lost more weight compared to the other risk group when engaging in physical activity that exceeds NIH.

Sailer et al. [35] examined subject’s response to exercise as it related to FTO genotype. Subjects underwent a 9-month lifestyle intervention that included 3 h of moderate physical exercise per week and monthly sessions with a dietary counselor. No significant differences in weight loss were observed in risk or no risk allele individuals. Interestingly, the greatest improvements in VO2Max were exhibited by the risk allele group. This study did not instruct subjects to reduce energy intake; therefore, it is not surprising that subjects did not lose a significant amount of weight [35]. The authors do not offer an explanation as to why the risk allele showed the greatest improvements in exercise capacity

Wang et al. prescribed an moderate aerobic training plan consisting of 200 min of weekly physical activity at 4–55% of the subject’s heart rate reserve. All subjects lost body mass. Further statistical analysis revealed that men with the risk allele experienced greater loss in body mass, BMI, muscle mass and lean body mass. These differences were not reported in female subjects. This study demonstrated the role of physical activity in weight management.

The aforementioned studies suggest that physical activity plays a role in regulating FTO gene expression. It is worth noting that both Wang et al. and Sailer et al. [35] reported greater response to exercise in risk allele groups. This seems counterintuitive because the propensity to be overweight or obese is attributed to the risk alleles. These studies demonstrate that risk alleles tend to have a higher BMI and body mass at baseline; thus, greater improvements could be attributed to a higher initial baseline body mass and/or BMI. Please refer to Table 3 for a summary of studies.

## 7. Conclusions

Nutritional genomics, or the way diet and genes interact, has the potential to prevent or treat disease through personalized recommendations based on an individual’s genes. The FTO gene has been identified as a potential risk factor for the etiology of obesity. There are several SNPs associated with the FTO gene. Nicknamed the “obesity gene”, individuals carrying one of the risk alleles are more likely to be overweight or obese compared to those without the risk. Research has identified clear differences in energy intake and eating behaviors attributed to specific FTO SNPs. The FTO gene and related behaviors may offer some insight into why some people gain weight while others do not. The literature shows that weight loss is possible when acute changes in diet and physical exercise are implemented. Numerous studies have highlighted different diets or exercise modalities; however, to date, there is not overwhelming evidence pointing at one diet or exercise intervention be superior to another.

## Figures and Tables

**Table 1 jfmk-07-00090-t001:** Common FTO variations.

SNP	Genotype	Description
rs9939609	TT	Normal Risk
	A/-	Increased risk for obesity (BMI ≥ 30) [2,4,9,10,11,12,13,14] Increased body mass. [4,9,12,15] Increased incidence of T2DM [9] Lower levels of satiety [10,16,17] Increased intake [13,15,18,19,20]
rs17817449	TT	Normal Risk
	A/G	Increased risk for obesity (BMI ≥ 30) [3,6,21] Preference for high fat foods [22] Increased intake of refined carbohydrates [20] Increased intake of saturated fats [20]
rs3751812	GG	Normal Risk
	TT	Increased intake Increased BMI [14]
rs1421085	TT	Normal Risk
	C/-	Increased levels of perceived hunger [22] Increased intake [23] Increased intake of refined carbohydrates [20] Increased intake of saturated fats [20] Increased risk for obesity (BMI ≥ 30) [3,24]
rs1121980	GG	Normal Risk
	A/ C/	Increased intake of refined carbohydrates [20] Increased intake of saturated fats [20]
rs8050136	CC	Normal Risk
	A/-	Increased risk for elevated BMI [14] Increased risk for T2DM [8]

**Table 2 jfmk-07-00090-t002:** Summary of Studies assessing Dietary Habits.

Study	Intervention/Assessment	Subject/Population	SNP	Outcomes?
Harbon et al. [22]	Baecke Questionnaire of Habitual Physical Activity Food frequency questionnaire (FFQ) Three-Factor Eating Questionnaire (TFEQ) General Health Questionnaire (GHQ) Beck Depression Inventory (BDI) Self-Esteem Scale (RSES)	25–40 yo BMI ≥ 27 kg/m^2^	rs17817449 rs1421085	Those with the risk alleles higher intake of high fat foods and refined starches and poorer eating habits.
Edmond et al. [10]	Child Eating Behavior Questionnaire Self-reported satiety (Likert Scale)	9–10 yo n = 178	rs9939609	Increased intake Increased BMI
McCaffery et al. [23]	Food Frequency Questionnaires	Adults BMI: 36.3 ± 6.1 n = 2075	rs1421085	Increased intake
Melhorn et al. [16]	Three-Factor Eating Questionnaire Revised Restraint Scale International Physical Activity Questionnaire Short Form	18–50 yo Twins BMI of 18.5–50 n = 62	rs9939609	Increased intake Low levels of satiety
Obregón et al. [15]	Eating in Absence of Hunger Questionnaire (EAHQ) The Child Eating Behavior Questionnaire (CEBQ) The Three Factor Eating Questionnaire (TFEQ-19) and The Food Reinforcement Value Questionnaire (FRVQ)	8–14 yo n = 253	rs9939609	Increased energy intake Increased body mass
Oyeyemi et al. [11]	Anthropometric assessment International Physical Activity Questionnaire-Short	17–39 yo n = 201	rs9939609	Increased BMI associated with risk allele Physical activity reduced likelihood of being overweight
Ranzenhofer et al. [18]	Anthropometric assessment Subjects consumed standardized breakfast followed by supervised lunch	5–10 yo n = 22	rs9939609	Increased caloric intake
Speakman et al. [19]	VO_2_ Max BMR Food diary	21–64 yo BMI: 16.7–49.3 kg/m^2^ n = 107	rs9939609	FTO risk allele influences food intake. Relationship between greater VO_2_ Max scores and lower BMI
Wardle et al. (2008) [12]	Child Eating Behavior Questionnaire (CEBQ) Satiety Responsiveness Scale Enjoyment of Food Anthropometric assessment	8.3–11.6 yo Twins n = 3337	rs9939609	Lower satiety and greater enjoyment for food in risk allele
Wardle et al. (2009) [17]	Intake in EAH test Maternal rating of child’s enjoyment of active pastimes Mother’s rating of child’s relative activity Fidgetiness Food intake in absence of hunger	Avg age 4.4 Twins n = 133	rs9939609	High consumption of highly palatable foods Lower levels of satiety No differences in fidgetiness and physical activity
Kim et al. [14]	Physical activity	7–85 yo Latino T2DM n = 667	rs3751812 rs8050136 rs9939609	Reduced obesity risk associated with physical activity Increased BMI associated with risk allele
Hubacek et al. [21]	Food frequency questionnaire (FFQ) BMR via predicted equation Self-reported physical activity	45–69 yo Caucasian n = 6024	rs17817449	Lower BMR in risk carriers
Zhang et al. [27]	Anthropometric assessment	30–70 yo BMI 25 to 40 kg/m^2^ n = 811	rs1558902	Greater weight loss in response to high protein diets in risk carriers
Nishida et al. [28]	Subjects randomly assigned to resistance training or non-resistance training group Anthropometric assessment 1-week dietary recall	Avg Age 50.6 ± 12.1 BMI of ≥25 kg/m^2^ Japanese females n = 18		
Steemburgo et al. [13]	3-day weighed diet record Metabolic labs BMR via predicted equation	Avg Age 60.0 ± 10.3 T2DM n = 236	rs9939609	Increased intake of fat, decreased fiber intake
Antonio et al. (2018)	Anthropometric assessment	Exercised-trained n = 108	rs1421085	Risk-allele carriers had higher fat mass and body fat percentage

**Table 3 jfmk-07-00090-t003:** Summary of intervention studies.

Study	Intervention		Subject/Population	SNP	Outcomes
Haupt et al. [30]	Nutrition	TULIP lifestyle intervention—reduced caloric intake from fat by 30%, increased fiber intake and perform 3 h of physical activity per week	German Inclusion: Family history of type 2 diabetes, a BMI > 27 kg/m^2^, or a previous diagnosis of impaired glucose tolerance or gestational diabetes n = 204	rs8050136	Risk allele was associated with increased BMI No statistical difference in weight loss between risk and non-risk allele
Antonio de Luis et al. [31]		3 month hypocaloric diet: Low-carbohydrate diet (38% carbohydrates, 26% proteins and 36% fats) Low-fat diet (53% carbohydrates, 20% proteins and 27% fats	n = 305 Obese	rs9939609	Non risk vs. risk Diet 1: Body mass (3.9 ± 3.4 vs. 3.2 ± 4.1 kg: *p* > 0.05) Fat mass, (1.7 ± 2.5 vs. 2.0 ± 4.0 kg; *p* > 0.05), waist circumference (3.5 ± 6.1 vs. 3.5 ± 6.7 cm; *p* > 0.05) Diet 2: weight (4.7 ± 4.1 vs. 3.9 ± 3.7 kg; *p* > 0.05), fat mass (3.8 ± 3.2 vs. 2.6 ± 3.7 kg; *p* > 0.05), waist circumference (5.8 ± 7.1 vs. 5.1 ± 5.3 cm; *p* > 0.05)
Razquin et al.		1 low-fat diet and 2 Mediterranean diets	n = 776 55–80 yo High cardiovascular risk	rs9939609	NEED HELP
Verhoef et al. [32]		Very low-calorie diet (<500 kcal) followed by 3 month weight maintenance	n = 150 20–50 y BMI of 27–38	rs9936909	No statistical differences in weight loss
Rauhio et al. [33]		3 month vey-low energy diet followed by 9 month maintenance diet	n = 75 BMI > 30 sedentary, premenopausal, women 25–45 y	rs9939609	No statistical differences between risk and non-risk allele in weight loss at 3 and 9 months
Antonio et al., (2019) [34]		4-week hypocaloric diet Anthropometric assessment	Exercise-trained n = 47	rs1421085 rs17817449 rs9939609	Short-term weight loss unaffected by FTO risk
Sailer et al. [35]	Exercise	9-month lifestyle intervention	Healthy adults n = 292	rs8050136	No statistical differences in weight loss in risk and non-risk carriers
Leońska-Duniec et al. [36]		12-week training program	n = 201 Polish women	rs9939609	No genotype or training interactions reported
Mitchell et al. [37]		6-month exercise training program	n = 464 45–75 y Female	rs8050136	Weight loss was greater in the A/A group All subjects improved cardiorespiratory fitness and lost weight
Wang et al.		12-week moderate aerobic exercise program	n = 240 21–60 y BMI: 28–40	rs8050136	Male subjects with risk allele lost more weight (−6.19 ± 6.62 vs. −4.18 ± 4.09, *p * = 0.008) No significant differences reported in females
Raniken et al. [38]		20-week exercise program	n = 276 Sedentary	rs8050136 rs9939609	No statistical differences in risk and non-risk allele in BMI and body mass. HELP
do Nascimento et al. [39]		After school exercise sessions 3 × week: land-based aerobic exercise, HIIT, combined training or water walking	n = 135 8–17 y Overweight or obese	rs9939609	
Nishida et al. [28]		Subjects randomly assigned to resistance training or non-resistance training group Anthropometric assessment 1-week dietary recall	Avg Age 50.6 ± 12.1 BMI of ≥25 kg/m^2^ Japanese females n = 18		

## Data Availability

Not applicable.

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
