# Peer review of "FTO and Anthropometrics: The Role of Modifiable Factors"

_jfmk, 2022, doi:10.3390/jfmk7040090_

Round 1

Reviewer 1 Report

Evans and colleagues reviewed the FTO single nucleotide polymorphism and its influence on risk for obesity. They selected original observational or interventional studies assessing  eating behaviors, eating preferences, nutrition interventions and/or other lifestyle factors for review. There are some issues to be corrected.

 The title of the manuscript, although intriguing and encouraging to read the article, does not fully reflect the content. The authors only analyze SNPs of the FTO gene, but do not mention other factors influencing the functioning of the gene, e.g. epigenetic changes. It is worth mentioning the influence of methylation and FTO gene expression on the possible development of obesity (e.g. https://doi.org/10.3390/nu13051683 or https://doi.org/10.3390/nu13051683)

The authors prepared structured review however there is no result section. Results seem to be presented in discussion. It should be moved to the result section.

Author Response

We appreciate the time and effort that the reviewers have devoted to reviewing our manuscript. We are grateful to the reviewers for their insightful comments on the paper. Please see below, in blue, for a point-by-point response to the reviewers’ comments and concerns.

The title of the manuscript, although intriguing and encouraging to read the article, does not fully reflect the content.

Thank you for the suggestion. We have updated the title to FTO and Anthropometrics: The Role of Modifiable Factors.

The authors only analyze SNPs of the FTO gene, but do not mention other factors influencing the functioning of the gene, e.g. epigenetic changes. It is worth mentioning the influence of methylation and FTO gene expression on the possible development of obesity (e.g. https://doi.org/10.3390/nu13051683 or https://doi.org/10.3390/nu13051683)

Thank you for the suggestion. We have added one sentence about factors influencing gene expression. This paper focuses on the correlation between FTO risk and lifestyle factors i.e. nutrition and physical activity. A more detailed explanation of epigenetics is beyond the scope of this paper.

The authors prepared a structured review however there is no result section. Results seem to be presented in the discussion. It should be moved to the result section.

Thank you for the suggestion. The paper is structured as a narrative review, which is intended to describe and appraise published studies. Unlike systematic reviews or meta-analyses, narrative reviews do not have results sections.

Reviewer 2 Report

Manuscript "FTO: Fat or Fiction?" presents an overview of the contemporary knowledge on the influence of genes on nutrition and nutritional disorders in humans and provides valuable information on the prevention of obesity and diet-related metabolic diseases.

Author Response

Thank you for your comments